# THE APPLICATION OF A GRAPH NEURAL NETWORK TO FORECAST THE RECIPROCAL IMPACT OF WELLS WITHIN AN OIL FIELD
# CONFERENCE SUBMISSIONS

**Yamkin M.A.**[*]**, Kugaevskikh A.V.**[†]
Department of Software Engineering
ITMO University
Saint Petersburg, Russia

**Koshkin T.A.** [‡]
Department of Integrated Asset Modeling
Engineering and technology service
Saint Petersburg, Russia

## ABSTRACT

The objective of this paper is to explore a novel approach for predicting the inter-dependencies between production and injection wells. The method under examination involves the application of graph neural networks.

The significance of this research stems from the necessity to understand the overall impact of enhanced oil recovery methods on field production, rather than focusing solely on the individual well where the activity is implemented. Currently, there is a lack of accurate, rapid, and cost-effective methods for identifying inter-well influences. Given the pressing nature of the challenge in determining mutual influences between wells, the authors have opted to address it through the utilization of graph neural networks.

The focus of this study is on various architectures of graph neural networks.

Research methodology involves the development of a graph neural network designed to predict the mutual influence between wells by extracting weights from the trained model.

As a result, several neural network architectures have been successfully trained, with the GraphSAGE architecture utilizing a long-short term memory (LSTM) aggregation function achieving the highest accuracy. The performance metrics for this neural network are as follows: r² - 0.97, MSE - 171.39, MAE - 5.46, RMSE - 13.09, based on an average oil flow rate of 272.33.

In conclusion, this study demonstrates the promising potential of employing graph neural networks for predicting the mutual influence between wells.

Keywords: Well interconnection, graph neural networks, GraphSAGE, LSTM, GGNN, oil flow rate.

## 1 INTRODUCTION

### 1.1 PURPOSE AND OBJECTIVES OF THE WORK

The objective of this paper is to examine a novel approach for predicting the mutual influence between production and injection wells, specifically through the application of graph neural networks.

To achieve this aim, the following objectives have been established:

1. Analyze existing methods for assessing well-to-well interference, identifying their respective advantages and disadvantages;

2. Represent the data in a graphical format and design a neural network architecture to address the problem;

---

[*]makson.yamkin@mail.ru

[†]avkugaevskikh@itmo.ru

[‡]koshkin.ta@gazprom-neft.ru

3. Determine the performance metrics of the developed neural network

## 1.2 RELEVANCE

Assessing the interaction between production and injection wells presents a significant challenge within the oil and gas sector. When implementing enhanced oil recovery methods, it is crucial to understand the overall impact on field production rather than solely on the individual well undergoing intervention. To ascertain this effect on the entire field, it is essential to comprehend the interconnections among the wells.

Currently, the following methods are employed to ascertain the degree of interconnectivity between wells:

1. Tracer studies. This method for determining the interdependencies between wells is elaborated upon in Raspopov A.V. (2022). The essence of tracer studies involves injecting fluid with indicators into injection wells. These indicators are transported to production wells through filtration flows, after which the relationship between production and injection wells is analyzed Raspopov A.V. (2022) . The primary disadvantages of this method include: high study costs and a lengthy duration (results are typically obtained after an average of six months) Raspopov A.V. (2022) ;

2. Hydro listening. This method for determining the relationship between production and injection wells is detailed in Gumerova A.A. (2022). The principle of hydro listening is to monitor pressure changes at wells when the operational mode of other wells is altered Gumerova A.A. (2022). The main drawbacks of this method are: high costs, challenges in interpreting study results, and low accuracy Gumerova A.A. (2022);

3. Statistical methods. These methods are described in detail in Romanenkov A.V. (2017), Khasanov et al. (2021), Tyrsin et al. (2023). The essence of statistical methods consists in application of hydrodynamic modelling and estimation of mutual influence between wells by means of analysis of field data Khasanov et al. (2021). Current method suffers from limitations such as high costs, lengthy analysis periods, interpretive difficulties, variable accuracy (ranging from 3 percent to 50 percent error), reliance on simplifying assumptions (e.g., invariant development systems), and the need for precise reservoir pressure data at all wells Romanenkov A.V. (2017). Furthermore, some methods are only applicable to individual well clusters, not the entire field Tyrsin et al. (2023).

4. Utilizing neural networks to ascertain the interconnections between wells presents a significant advancement Zhao-Qin et al. (2024), Leding et al. (2022). However, the primary drawback of the currently employed methodologies lies in their reliance on available data regarding the filtration and capacitance characteristics of the reservoir, which are often incomplete and unreliable. Furthermore, these approaches fail to incorporate the temporal aspect essential for accurately predicting the mutual influence between wells, a critical factor when analyzing field data. In addition, some of the modern research is based on tracer studies and petrophysical research, which are not always carried out at the field and not cost-effective. All of these are significant drawbacks.

Given the absence of a precise and cost-effective method for assessing the interrelationships among wells, particularly in the context of the oil and gas industry, the authors have undertaken a thorough analysis of this issue. In this study, the authors explore the application of graph neural networks to forecast the reciprocal influence between wells. Throughout the course of this research, the authors have developed several innovative neural network architectures.

## 2 RESEARCH METHODS

### 2.1 PROBLEM STATEMENT

The target variable, which represents the coefficient of mutual influence between wells, is absent in the initial dataset. Consequently, the designed neural networks were trained to predict an indirect parameter: the oil flow rate of producing wells.

By training the neural network to predict this parameter, it will acquire sufficient knowledge regarding the graph properties and its topology. Upon completion of the training, the weights of the neural network will be extracted, serving as the coefficients of mutual influence between wells, grounded

in the theory of graph neural networks Sun et al. (2024), Groh et al. (2023), Kuo et al. (2024). These coefficients will illustrate the extent to which the graph nodes (wells) influence one another Sun et al. (2024), Groh et al. (2023), Kuo et al. (2024).

## 2.2 FEATURES

The authors utilized field data from the N field, situated in Western Siberia, to evaluate the approach of predicting mutual influence between wells through neural networks. The reservoir of this field is terrigenous. A specific section of this field was selected for testing, where 28 wells were drilled.

The initial data were represented in the form of a graph: the nodes of the graph correspond to wells, with each node characterized by a specific embedding—a set of features for each well (the nodes of the graph will be of two types, as there are both producing and injection wells at the site in question); the edges of the graph represent connections between wells. For the initial approximation, the wells are considered to be strongly connected—meaning all wells are interconnected (producing wells to producing wells, injection wells to injection wells, and producing wells to injection wells)—as all wells exert influence on one another Romanenkov A.V. (2017), Khasanov et al. (2021), Tyrsin et al. (2023). A graph was generated for each date present in the training sample, with data provided for the years 2020-2024. A total of 1821 graphs were included in the dataset, reflecting the number of wells in the field on a given date. Each graph includes all wells and their embeddings for different time periods. The training and test samples were divided in a 90:10 ratio.

The initial data are fed into a graph neural network, which predicts the oil flow rate for each node representing a producing well. Subsequently, weights are extracted from the trained neural network, which serve as coefficients characterizing the degree of mutual influence between wells. The calculated coefficients are assigned to each edge of the graph.

The following features were selected to train the graph neural network:

1. For production wells: water cut, electric submersible pump frequency, bottomhole pressure, reservoir pressure, gas flow rate, and wellhead pressure. The target attribute is the oil flow rate;
2. For injection wells: wellhead pressure and injected fluid flow rate. The target attribute is not applicable for these wells, as they do not produce oil.

Additionally, a common attribute for all wells was utilized: well type.

These data were selected due to their frequent analysis by oil and gas specialists when examining the mutual influence between wells Romanenkov A.V. (2017), Khasanov et al. (2021), Tyrsin et al. (2023).

## 2.3 NEURAL NETWORK ARCHITECTURES

To achieve the objectives of this paper, various graph neural network architectures have been explored.

Given that oil field data is dynamic, i.e., it changes over time, it is essential to employ graph neural networks that account for the temporal component.

### 2.3.1 TEMPORAL GRAPH CONVOLUTIONAL NETWORKS

Temporal Graph Convolutional Networks (TGCN) facilitate the processing of a graph by utilizing aggregated information from the neighbors of each node Sun et al. (2024). This network comprises two components: Graph Convolutional Networks (GCN)—a graph neural network that processes the graph—and Gated Recurrent Unit (GRU)—a recurrent neural network that models temporal dynamics.

### 2.3.2 GATED GRAPH NEURAL NETWORK

Gated Graph Neural Network (GGNN) also integrates the principles of recurrent and graph neural networks Groh et al. (2023). In the initial stage, information from the neighbors of each node is aggregated using convolutions Groh et al. (2023). Subsequently, the state of each node is updated using GRU, which allows for dynamic storage or forgetting of information Groh et al. (2023).

### 2.3.3 GRAPHSAGE

This network facilitates the modification of node embeddings by employing an aggregation function that considers a specified number of neighbors. The following functions may serve as aggregation methods: mean, long-short term memory (LSTM), or pooling Kuo et al. (2024). Given that the task involves time series analysis, it is imperative to utilize LSTM aggregation.

### 2.3.4 GRAPH ATTENTION NETWORKS

Graph Attention Networks (GAT) leverage attention mechanisms to identify the most significant neighbors, thereby circumventing the processing of redundant information Zhao et al. (2021). An additional recurrent layer, LSTM, is incorporated to account for the temporal component.

## 2.4 EXTRACTING WEIGHTS FROM THE MODEL

Upon the completion of training for each neural network, the weights were extracted from the neural networks and placed in correspondence with the edges of the graph.

## 3 RESULTS OF THE STUDY

### 3.1 FEATURE PROCESSING

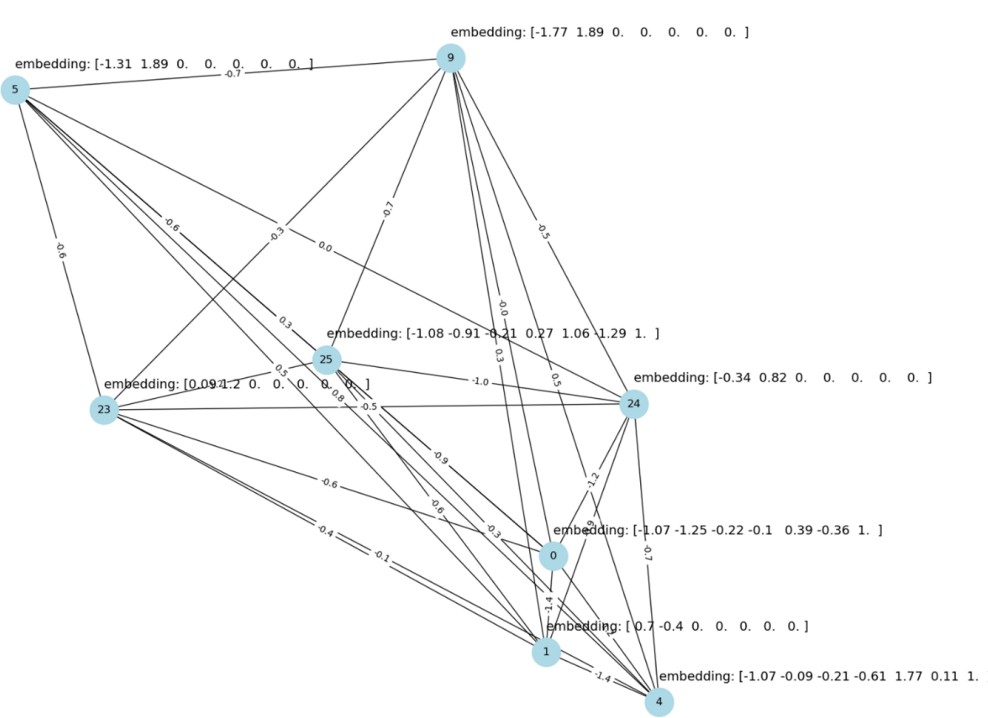

Figure 1: Graphical representation of the northern section of the field. Blue circles denote wells; the edges signify interconnections. The number on each edge indicates the distance between the wells. The "embedding" field encompasses the features selected for training the model.

To visualize the data upon which the neural network is trained, the authors has crafted a graphical representation illustrating the northern section of the field (Figure 1).

Table 1: Summary of used neural network architectures

| Architecture | Number of epochs | Learning rate |
|---|---|---|
| GraphSAGE | 180 | 200 |
| TGAT | 200 | 200 |
| GGNN | 150 | 200 |
| TGCN | 800 | 200 |

Table 2: Metrics of used neural network architectures

| Architecture | r² | MSE | MAE | RMSE |
|---|---|---|---|---|
| GraphSAGE | 0.97 | 171.39 | 5.46 | 13.09 |
| TGAT | 0.88 | 759.18 | 12.30 | 27.55 |
| GGNN | 0.96 | 215.49 | 5.84 | 14.68 |
| TGCN | 0.31 | 2501.09 | 33.16 | 50.01 |

## 3.2 TRAINING OF NEURAL NETWORKS

A total of four neural network architectures were trained: GraphSAGE, TGAT, GGNN, and TGCN. For each architecture, the mean squared error (MSE) function was employed as the loss function, given that the problem at hand is a regression issue. The Adam optimizer was utilized.

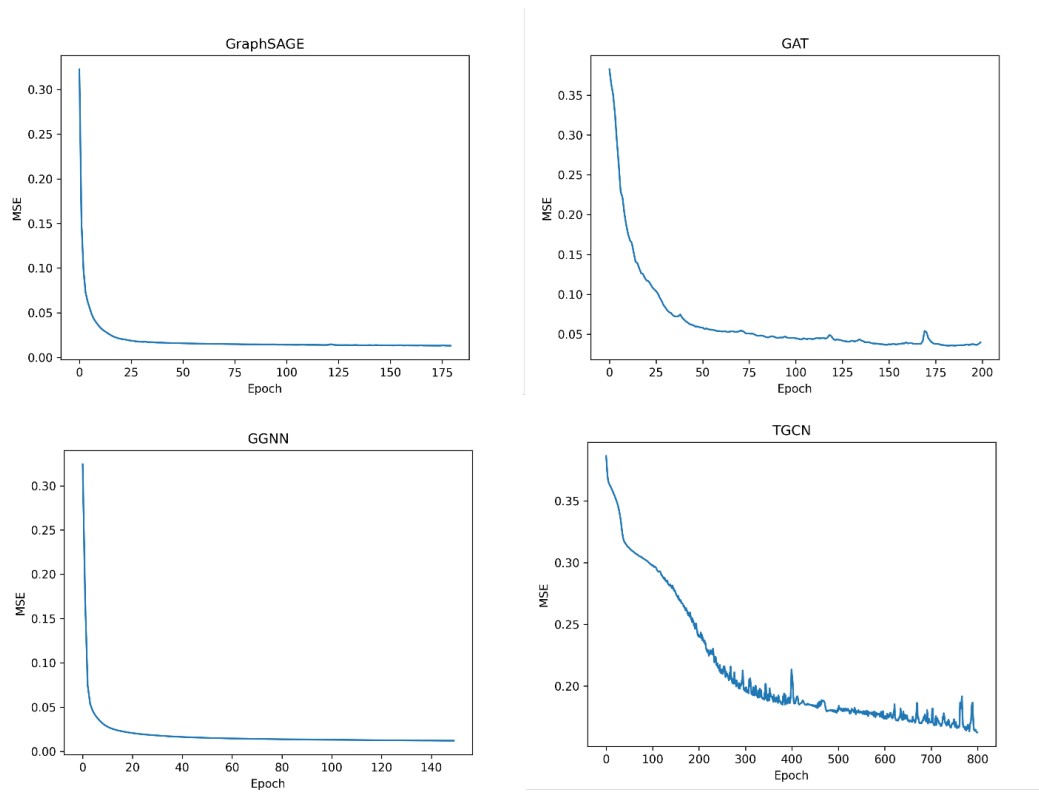

Figure 2: Loss function for each neural network architecture.

Figure 2 shows the plots of the loss function for each network.

Table 1 summarizes the fundamental information regarding the neural network architectures employed.

Table 2 presents the metric values on the test sample for each architecture. The following metrics were selected for accuracy assessment: mean absolute error (MAE), MSE, root mean squared error (RMSE), and r². These metrics were chosen due to their prevalence in evaluating the accuracy of regression problems.

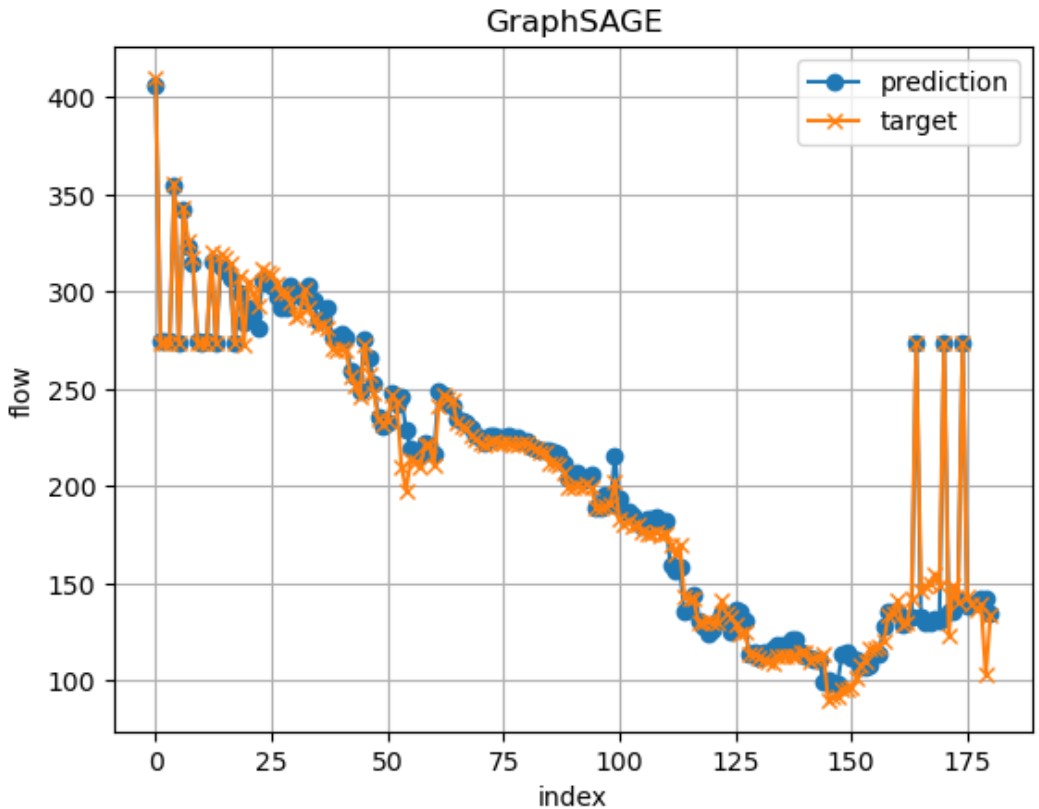

Figure 3: Correspondence of predicted and actual flow rates (GraphSAGE).

It is evident from the table that the GraphSAGE neural network architecture demonstrates the highest accuracy. The average flow rate for all wells is 272.33. For enhanced visualization, the authors provide graphs illustrating the correlation between the predicted flow rates and the actual ones using the GraphSAGE and GGNN models (Figure 3-4).

## 4    DISCUSSION OF THE RESULTS OF THE STUDY

### 4.1    FEATURES

During the feature selection process, the authors conducted heatmap analysis for producer and injector wells (Figure 5-6) and feature importance analysis (Figure 7). This was performed to eliminate unnecessary features prior to training if they exhibited high correlation, and post-training if they proved unimportant for model training. No importance analysis was conducted for injection wells, as they lack a target feature, specifically the oil flow rate. The mapping of feature names in the model for training is illustrated in Table 3.

Other attributes may be utilized in the future. Given that, based on Figure 4, the gas flow rate attribute holds significant importance, it may be feasible to substitute the water cut attribute with the water flow rate.

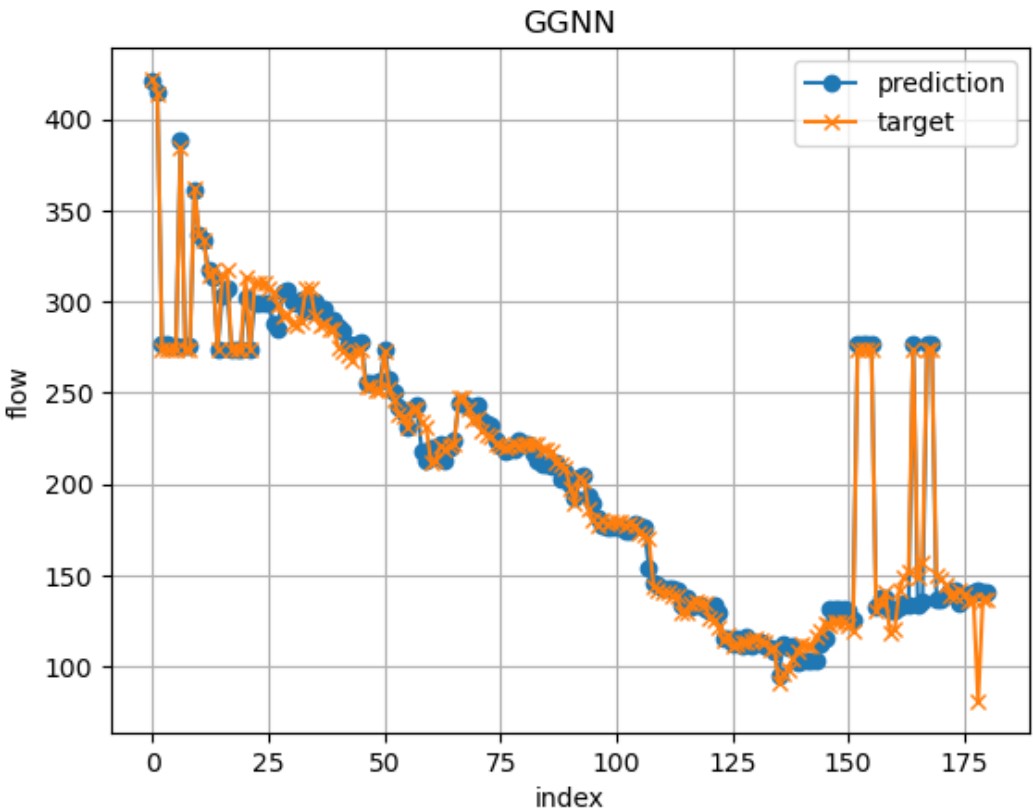

Figure 4: Correspondence of predicted and actual flow rates (GGNN).

Table 3: Summary of used neural network architectures

| Feature | Feature in model |
| --- | --- |
| Wellhead pressure | phead |
| Gas flow rate | gasflow |
| Reservoir pressure | pplast |
| Downhole pressure | pdown |
| Electrical submersible pump pressure | freq |
| Water cut | watercut |
| Oil flow rate / injected flow rate | flow |
| Type of well | welltype |

Additionally, in certain columns of the initial data, the features were populated using the following methods: linearization (thus filling in the gaps in the water cut feature) and carrying forward the first value (thereby addressing the gaps in the wellhead pressure, frequency, and bottomhole pressure features). These methods of feature imputation are employed by specialists in oil and gas companies, which is why they were selected Orlova et al. (2023).

In this task, a graph with two types of nodes was processed: production and injection wells. Each type of node possesses a distinct number of features, necessitating specialized processing. For the initial approximation, the missing values in the embeddings of the injection wells were filled with zeros. Since these values will not be processed by the neural network (multiplication by zero yields no result), this method is deemed feasible. However, there are plans to implement alternative pro-

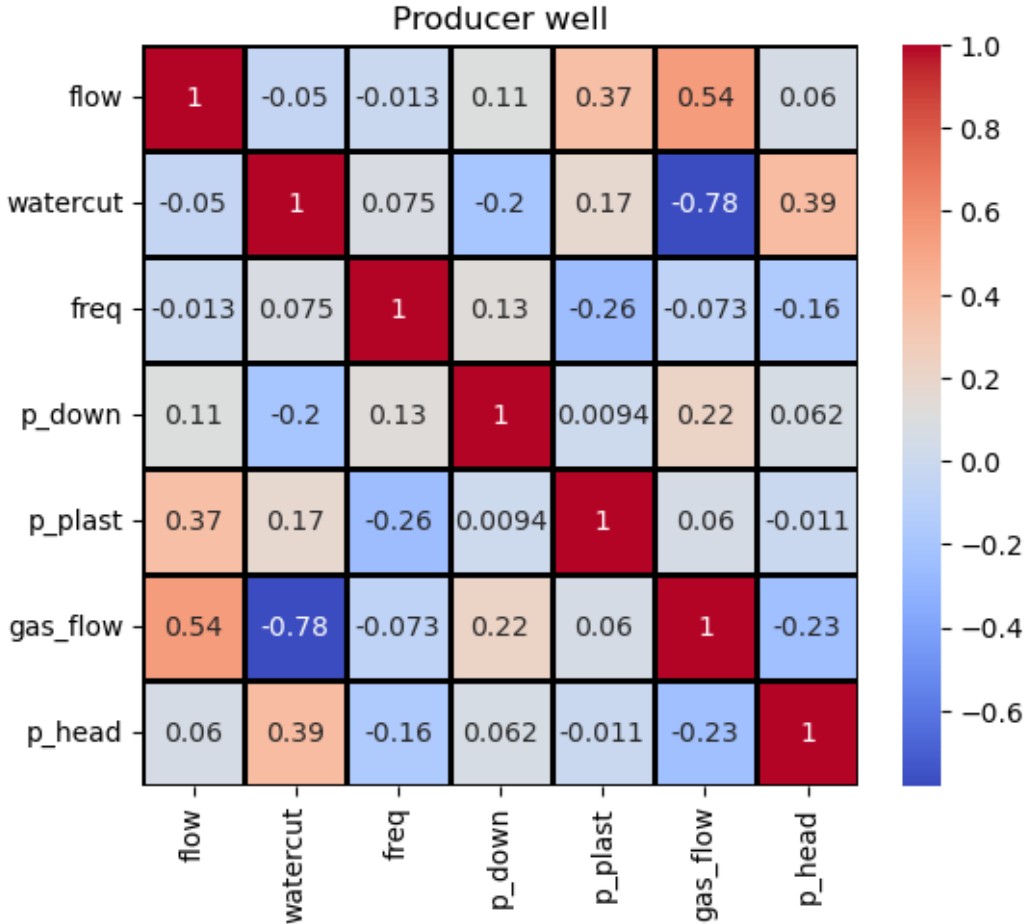

Figure 5: Heatmap analysis of production wells.

cessing methods for different types of nodes in the graph in the future, such as encoding with a Variational Autoencoder (VAE) neural network or a Multilayer Perceptron (MLP) neural network.

All features were normalized using the StandardScaler function from the sklearn library. In the future, it may be possible to explore other normalization functions or to conduct model training without normalization.

## 4.2 NEURAL NETWORK ARCHITECTURES

The authors implemented training using batches, with a batch size of 40. A test sample comprising 10 percent of the entire dataset was designated to assess the presence or absence of overfitting. All metric values presented in Table 2 were derived from the test dataset, confirming that overfitting is absent across all models.

Simultaneously, based on the loss function plots (Figure 2) and the metric values from the test dataset (Table 2), the authors conclude that the TGCN model failed to adequately approximate the data. This network does not account for the significance of neighboring nodes, in contrast to other architectures such as GraphSAGE and its counterparts. This oversight may lead to the processing of residual information, resulting in the network's inability to approximate the data effectively.

In contrast, the other neural network architectures (GraphSAGE, TGAT, GGNN) demonstrated commendable performance, as evidenced by the favorable metric values in Table 2. The GraphSAGE

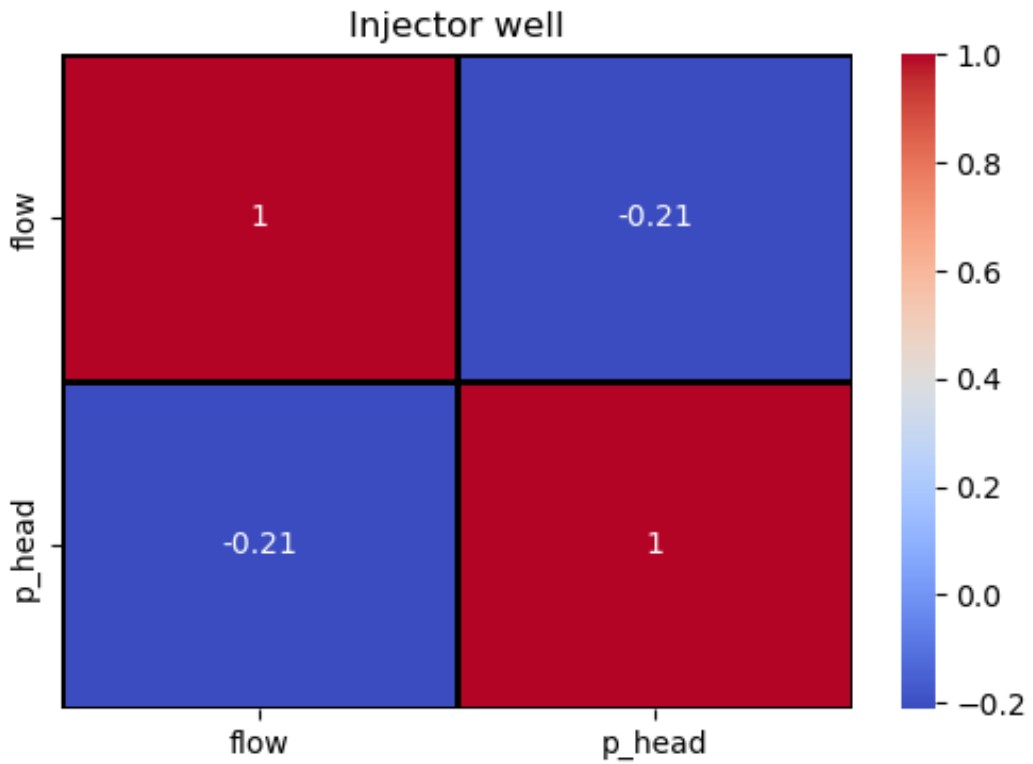

Figure 6: Heatmap analysis of injection wells.

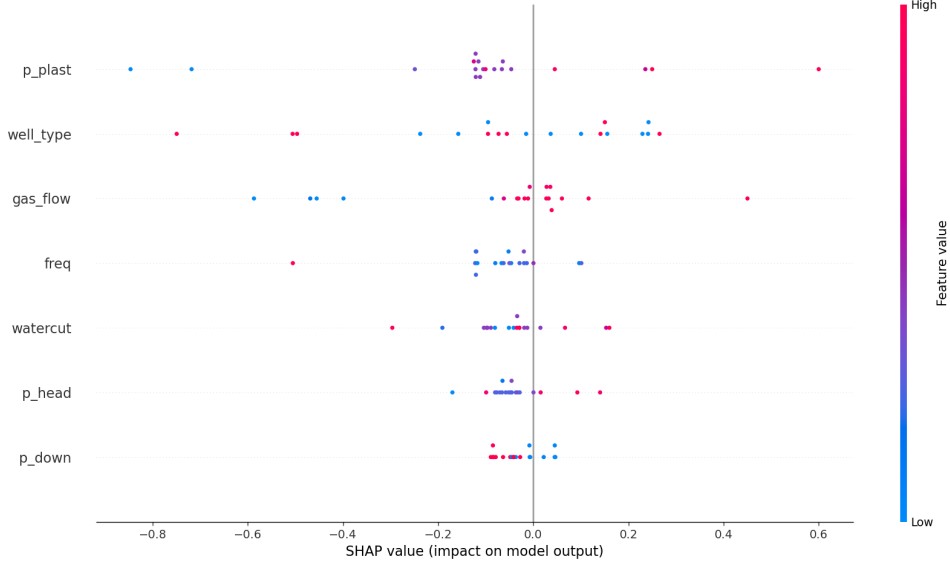

Figure 7: Analysis of the importance of the features of producing wells.

model achieved the highest metric values, indicating that this architecture is the most promising for future applications.

Looking ahead, the authors intend to explore additional neural network architectures to enhance prediction accuracy. For instance, the Spatio-Temporal GAT (ST-GAT) architecture, which considers both the temporal dynamics of graph data and the spatial positioning of graph nodes, shows significant promise.

### 4.3  DETERMINATION OF MUTUAL INFLUENCE COEFFICIENTS

The trained neural network accurately predicts the production well flow rate, indicating that it has effectively discerned the graph topology and its properties.

The decision to predict well flow rate was made because it serves as the primary parameter for assessing whether changes in a well are influenced by neighboring wells Raspopov A.V. (2022), Gumerova A.A. (2022), Romanenkov A.V. (2017). Consequently, variations in flow rate can be utilized to identify the factors affecting it (e.g., changes in flow rate at an injection well or an increase in withdrawal rate at another production well) Raspopov A.V. (2022), Gumerova A.A. (2022), Romanenkov A.V. (2017).

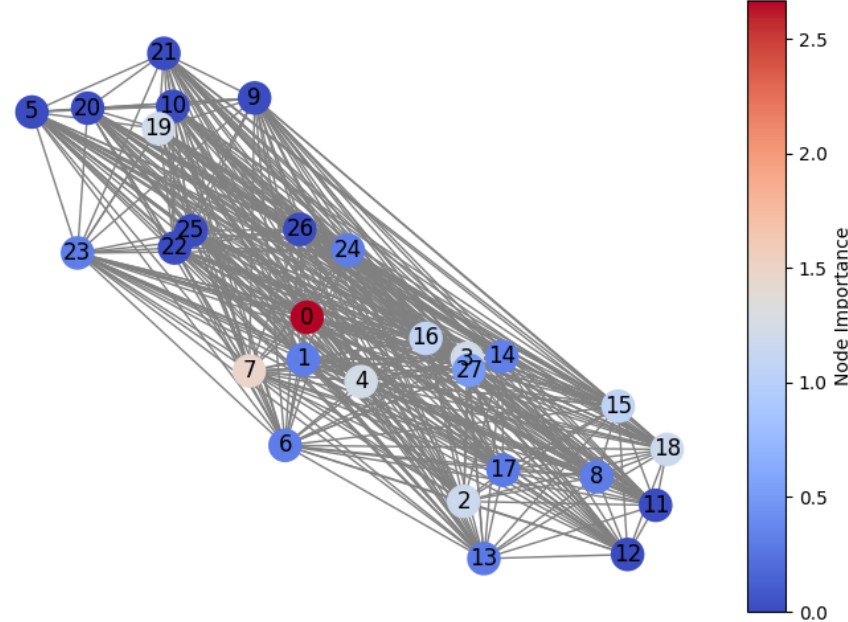

Figure 8: Analysis of the reciprocal impact of wells (wells that influence well number 0).

After training all models, weights were extracted and assigned to the edges. All weights were normalized using the MinMaxScaler function from the sklearn library to scale the mutual influence coefficients between 0 and 1. As previously described, these extracted weights can be referred to as mutual influence coefficients, as they reflect the significance of neighboring nodes for each graph node. Consequently, for each well, it is feasible to identify the other wells that exert the greatest influence upon it (Figure 8). Given that the neural networks (GraphSAGE, TGAT, GGNN) exhibit high metric values, their accuracy in predicting mutual influence between wells is correspondingly high.

Each neural network architecture comprised several layers. Weights were extracted from the first layer of each network, as this information is the least generalized. In contrast, the last layer aggregates information from the entire graph at each node, which is inappropriate since nodes that are more distant from the node in question should not be considered.

## 5 CONCLUSION

The study successfully achieved all its objectives:

1. A comprehensive analysis of current approaches to identifying well-to-well interference was conducted, revealing the relevance of the problem due to the lack of an accurate, cost-effective, and rapid method for determining mutual influence.

2. The data was formatted appropriately for the problem, leading to the design of four neural network architectures, each tailored to the specifics of the problem, including nodes of varying types and the dynamic nature of the graph.

3. The GraphSAGE neural network exhibited the highest metric values, with the following metrics: $r^2$ - 0.97, MSE - 171.39, MAE - 5.46, RMSE - 13.09, based on an average oil flow rate of 272.33.

Weights were extracted from each neural network, representing the coefficients of mutual influence between wells. These extracted weights demonstrated high accuracy when compared to the actual values of the mutual influence coefficients.

The approach examined for predicting the interdependence between wells shows great promise for practical application. This method is distinguished by its high accuracy and efficiency. It now enables the assessment of how an enhanced oil recovery operation at one well will impact other wells and the overall production from the entire field.

Looking ahead, there are plans to refine the application of neural networks for predicting the mutual influence between wells by exploring alternative feature processing techniques and different neural network architectures.

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
