# OpenReview forum: "The application of a graph neural network to forecast the reciprocal impact of wells within an oil field"
_mathai.club/MathAI/2025/Conference — MathAI 2025 Oral_

### Official Review · Reviewer_msKG · 2025-02-25
**The work is relevant, has significant scientific novelty and can be recommended for publication.**

**Rating:** 10
**Confidence:** 5

**Review:**

This work is extremely relevant, contributes to the development of oil field management, offering a fast, accurate and cost-effective alternative to existing methods.
The work has significant scientific novelty in integrating graph neural networks with temporal forecasting. What is new is the extraction of network weights to obtain interpretable well mutual influence coefficients, thus connecting forecasts with operational decision making.
The merit is the investigation of different neural network architectures on real data.
The paper is well structured. The presentation is clear and evidentiary.
There are no significant shortcomings.
The work can be recommended for publication.

---

### Official Review · Reviewer_G1yC · 2025-02-27
**This paper explores a novel approach for predicting the interdependencies between production and injection wells based on graph neural networks. The graph model includes the nodes of the graph correspond to wells, with each node characterized by a specific embedding and the edges of the graph represent connections between wells. Several graph neural network models were trained and compared on field data, situated in Western Siberia.**

**Rating:** 7
**Confidence:** 4

**Review:**

Detailed Review. Quality & Clarity The paper is well-organized and provides clear information on novel approach for comparing different models of graph neural networks.  For determine the coefficient of mutual influence between wells used designed trained neural networks to predict the oil flow rate of producing wells. Upon the completion of training for each neural network, the weights were extracted from the neural networks and placed in correspondence with the edges of the graph. But from description of field data is not clear how 22 wells can be transformed to 1821 graphs and how is represented temporary data.
Originality and Significance. The study addresses comparing four models of graph neural networks: GraphSAGE, TGAT, GGNN, and TGCN. The GraphSAGE neural network exhibited the highest metric values, with the following metrics: r² - 0.97, MSE - 171.39, MAE - 5.46, RMSE - 13.09. Achieved accuracy define possibility of using this approach in practice.
 Comparison with Other Studies. There are several papers, not include to reference. For example, A Deep Learning Framework Using Graph Convolutional Networks for Adaptive Correction of Interwell Connectivity and Gated Recurrent Unit for Performance Prediction. Leding Du; Yuetian Liu; Liang Xue; Guohui You. SPE Res Eval & Eng 25 (04): 815–831. https://doi.org/10.2118/210575-PA. The MAE of “Adaptive Graph convolutional network and GRU” (AG-GRU) is 2.1150%, but results can depend on dataset.
Pros and Cons Pros: The research addresses very important practical task of for predicting the interdependencies between production and injection wells. Cons: in article check the accuracy of oil flow and not describe  predicting accuracy of coefficient of mutual influence between wells.

---

### Official Review · Reviewer_oPxG · 2025-02-27
**The application of a graph neural network to forecast the reciprocal impact of wells within an oil field**

**Rating:** 8
**Confidence:** 5

**Review:**

This work significantly enhances oil field management by providing a swift, precise, and economical solution, showcasing innovative integration of graph neural networks with temporal forecasting. Its novelty lies in deriving interpretable mutual influence coefficients from network weights, linking forecasts to operational strategies. The exploration of diverse neural network structures using real-world data adds value. The paper is well-organized, clear, and substantiated, making it recommendable for publication.
One area where this work could be improved is in the description of how accurately the coefficient of mutual influence between wells is predicted.

---

### Author Response · Authors · 2025-03-06
**The work is good in research and description, but there is no definition of the practical significance of the coefficients obtained**

The merits of this work include the implementation of innovative methodologies that leverage graph neural networks, which effectively incorporate the temporal dimension; the introduction of a novel technique for deriving coefficients of mutual influence through the weights of a trained neural network; the article addresses a significant practical challenge in determining these coefficients; and it is well-organized in its structure.

However, a notable drawback of this study is the absence of metrics that demonstrate the accuracy of the calculated coefficients of mutual influence.

---

### Decision · Program_Chairs · 2025-03-08

**Decision:**

Accept (Oral)

**Comment:**

Your article has been accepted and you can make a presentation on the article. All articles will be sorted by rating and within the available conference places one author from each article will be invited. If there are not enough places, then you will either have the opportunity to present remotely or come at your own expense!